

# Constraining 2010-2020 Amazonian carbon flux estimates with satellite solar-induced fluorescence (SIF)

Archana Dayalu[1], Marikate Mountain[1], Bharat Rastogi[2,3,4], John B. Miller[3], Luciana Gatti[5]

[1]Verisk Atmospheric and Environmental Research, Lexington, 02420, USA
[2]Cooperative Institute for Research in Environmental Sciences, University of Colorado Boulder, Boulder 80309, USA
[3]NOAA Global Monitoring Laboratory, Boulder, 80303, USA
[4]Department of Geography, University of Colorado Boulder, Boulder 80309, USA * present affiliation
[5]National Space Research Institute (INPE) LaGEE Greenhouse gas laboratory, Sao Jose dos Campos, 12227-010, Brazil

*Correspondence to*: Archana Dayalu (adayalu@aer.com)

**Abstract.** Amazonia's Net Biome Exchange (NBE), the sum of biogenic and wildfire carbon fluxes, is a fundamental indicator of the state of its ecosystems. It also quantifies the magnitude and patterns of short- and long-term carbon dioxide sources and sinks but is poorly quantified and out of equilibrium (non-zero) due to both direct (deforestation) and indirect (climate-related) anthropogenic disturbance. Determining trends in Amazonia's carbon balance, shifts in carbon exchange pathways of NBE, and timescales of ecosystem sensitivity to disturbance requires reliable biogenic flux models that adequately
capture fluxes from diurnal to seasonal and annual timescales. Our study assimilates readily available observations and a derived solar-induced fluorescence (SIF) product to estimate hourly biogenic carbon dioxide ($CO_2$) fluxes (here in units of $\mu mol\ CO_2\ m^{-2}\ s^{-1}$) as Net Ecosystem Exchange (NEE), and its photosynthesis and respiration constituents, at 12 km resolution using four versions of the data-driven diagnostic Vegetation Photosynthesis and Respiration Model (VPRM). The VPRM versions are all calibrated with ground-based eddy flux data and vary based on whether (1) the photosynthesis term incorporates
SIF (VPRM_SIF) or traditional surface reflectance (VPRM_TRA) and (2) the respiration term is modified beyond a simple linear air temperature dependence (VPRM_SIFg; VPRM_TRG). We compare the VPRM versions with each other and with hourly fluxes from the bottom-up mechanistic Simple Biosphere 4 (SiB4 v4.2) model. We also use NASA's OCO-2 $CO_2$ column observations to optimize the VPRM and SiB4 models during the 2016 wet season which occurred at the tail of the 2015/2016 severe El Niño. The wet season 2016 case study suggests that relative to SiB4 and the SIF-based VPRMs, the
traditional VPRM versions can underestimate uptake by a factor of three. In addition, the VPRM_SIFg version better captures biogenic $CO_2$ fluxes at hourly to seasonal scales than all other VPRM versions in both anomalously wet and anomalously dry conditions. We also find that the VPRM_SIFg model and the independent bottom-up mechanistic hourly SiB4 model converge in NEE, although there are differences in the partitioning of the photosynthesis and respiration components. We further note that VPRM_SIFg describes greater spatial heterogeneity in carbon exchange throughout the Amazon. Despite the paucity of
OCO-2 $CO_2$ column observations ($XCO_2$) over the Amazon in the wet season, incorporating $XCO_2$ into the models significantly reduces near-field model-measurement mismatch at aircraft vertical profiling locations. Finally, a qualitative analysis of the unoptimized biogenic models from 2010-2020 agrees with the wet season 2016 case study, where the traditional



VPRM formulations significantly underestimate photosynthesis and respiration relative to VPRM-SIFg. Overall, the VPRM_SIFg biogenic flux model shows promise in its ability to capture Amazonian carbon fluxes across multiple timescale and moisture regimes, suggesting its suitability for larger studies evaluating interannual and seasonal carbon trends in fire as well as the biogenic components of the region's NBE.

## 1 Introduction

The terrestrial tropics dominate the interannual variability of the global carbon cycle (Piao et al., 2020). As the dominant component of the South American terrestrial tropics, the Amazon is a major contributor to both the long-term global terrestrial carbon balance and the inter-annual variations in the terrestrial biosphere's sequestration of anthropogenic $CO_2$ emissions (Phillips & Brienen, 2017; Gatti et al., 2014; Saleska et al., 2003). Covering about one-third of South America with about 80% as tropical rain forest, the tropical Amazon is being altered profoundly by human activity especially in recent decades (Andreae, 2019; Fu et al., 2013). Gatti et al. (2021) found that, from 2010–2018, most of the Amazon's carbon emissions come from the disturbed eastern Amazon, and most of those emissions are from fires. In the past two decades, 44% of carbon loss from the Brazilian Amazon has been attributed to degradation and disturbance, and 56% to deforestation (Kruid et al., 2021). Recent work (Aragão et al. 2018) suggests that drought-related fires are increasing in importance relative to deforestation-related fires, but this statement was made under the assumption that protections of the Amazon against deforestation would remain in place. Following over a decade of decline, swaths of the Amazon have been illegally burned beginning with the 2019 dry-season; there was a three-fold increase in fire activity in 2019 compared to the previous year (Brando et al., 2020). Furthermore, the fire severity is potentially exacerbated by increased ecosystem fragmentation: aggregate impacts of human activity reduce the buffer that would have been present in a healthier and more intact ecosystem, making the Amazon more vulnerable to fire impacts (Fu et al., 2013; Aragão et al., 2018; Alden et al., 2016; Brando et al., 2020).

The associated carbon response to large scale fire activity is complicated given that the Amazon is not evolutionarily adapted to fires: while fires have played a historical role in the region, they have almost all been started by humans, generally for agricultural purposes (Uhl et al., 1988; Brando et al., 2020). As such, it is uncertain how prior burn trauma impacts future ecosystem productivity (Trumbore et al., 2015). Yet the direct and indirect carbon effects of the fires ravaging the Amazon during the dry seasons, particularly along the south-eastern frontier termed the "Arc of Deforestation" (Fig. 1), are playing an increasing role in the region's Net Biome Exchange (NBE) (Gatti et al., 2021). With the dry-season length of southern Amazonia already increasing significantly by approximately 6.5 days per decade, conditions conducive to enhanced fire seasons are further prolonged (Fu et al., 2013). Therefore, we are at a critical time for both the Amazonian and global carbon balance where a warmer, drier Amazon is poised to face additional pressures from the combined effects of both drought- and deforestation-related fires (Brando et al., 2020; Naus et al., 2022). Furthermore, under Brazil's volatile political regime, any re-established protections face an uncertain future (Brando et al., 2020; Naus et al., 2022; Gatti et al., 2023).



Despite the carbon response of the Amazon being a key variable in climate modeling (e.g., Cox et al., 2000, 2013), regional surface flux estimates of $CO_2$ as well as the atmospheric measurements to constrain them have been historically sparse due to physical, economic, and political limitations. As a result, capturing Amazon carbon surface fluxes and their sensitivities has been a challenge (Liu et al., 2017). However, the past two decades have seen an increase in multi-platform measurement efforts with the potential to greatly improve our ability to quantify carbon cycling in the Amazonian biosphere. Ground-based carbon flux measurements through AmeriFlux and the Amazon Tall Tower Observatory (ATTO) provide important surface-based ecosystem information at a high temporal resolution (e.g., Basso et al., 2023; Paca et al., 2022; Hayek et al., 2018). Aircraft observations have provided important constraints enabling in-depth assessments of land-atmosphere carbon exchange (e.g., Gatti et al., 2010, 2014, 2021; Gloor et al., 2012; van der Laan-Luijkx et al., 2015; Alden et al., 2016). Beginning in 2010, sustained semimonthly aircraft measurements have been conducted in the four corners of the Brazilian Amazon as an international effort among researchers, informing estimates of fire effects in the region (Gatti et al., 2021). Finally, total column $CO_2$ retrievals and solar-induced fluorescence (SIF) from NASA's Orbiting Carbon Observatory (OCO-2; operational September 2014) enable previously inferred features of the tropical biosphere's carbon response to climatological variability to be more accurately estimated at multiple spatiotemporal scales (Bloom et al., 2016).

With the Amazon becoming warmer and drier, the impacts of fires and other disturbances—both climate-related and directly through deforestation and other degradation—on carbon stock loss are compounded (Barkhordian et al., 2019; Kruid et al., 2021). However, absent a consistent observationally constrained multi-decadal record extending into the present, where $CO_2$ surface fluxes from both photosynthesis and respiration are captured, our capacity to evaluate long-term shifts in carbon dynamics of the South American tropics is limited. Existing global flux products with multi-decadal temporal records such as Gross Primary Productivity (GPP) estimates from NASA's MODIS have been found to inadequately represent complex Amazonian ecosystems (Almeida et al., 2018). In recent years, the availability of global satellite-based SIF observations—a more direct correlate to photosynthetic activity, including in the Amazon—has greatly improved our ability to capture the timing and magnitude of ecosystem carbon exchange (e.g., Doughty et al., 2019; Koren et al., 2018; Zhang et al. 2018a; Luus et al. 2017). To this end, our study uses SIF along with a suite of readily available ground-to-satellite observations to estimate hourly biogenic $CO_2$ fluxes ($\mu$mol $CO_2$ m$^{-2}$ s$^{-1}$) via an improved version of the Vegetation, Photosynthesis, and Respiration model (VPRM) (Mahadevan et al., 2008). The VPRM is a minimally parameterized data-driven diagnostic ecosystem light-use efficiency model that has many advantages for exploring shifts in regional carbon dynamics and modes of carbon exchange, particularly in its translation of local ecosystem and satellite observations into spatially resolved fluxes of $CO_2$ as Net Ecosystem Exchange (NEE), GPP, and $R_{eco}$ at fine spatial (here, 12 km) and temporal (hourly) scales. The VPRM has demonstrated effectiveness as a prior in diverse $CO_2$ source attribution studies at midlatitudes and the Arctic (e.g., Dayalu et al., 2018; Luus et al. 2017; Hilton et al., 2013; Matross et al., 2006). Furthermore, the region-specific approach of the VPRM contrasts with subsets of existing global vegetation carbon flux models which, in data sparse regions such as the Amazon, can be insufficiently resolved (Dayalu et al., 2018; Luus et al., 2017; Hilton et al., 2013; Matross et al., 2006). The VPRM is calibrated with direct ecosystem observations when available for each land use category in the domain of interest. Traditionally,



satellite surface reflectance-based proxies such as the Enhanced Vegetation Index (EVI) and the Land Surface Water Index (LSWI) are incorporated into the VPRM; however, there are known issues with that approach, particularly pertaining to LSWI
inadequately capturing the effects of water stress (Dayalu et al., 2018; Luus et al., 2017).

The SIF-based formulation of the VPRM has improved performance relative to the traditional VPRM (VPRM-TRA) when compared against aircraft and tower observations in extratropical settings (Luus et al., 2017). However, the SIF-based VPRM holds promise for the tropics as well: SIF is implicitly and significantly correlated to previously hard-to-quantify parameters like water stress (e.g., Mohammadi et al., 2022). Particularly in the Amazon, where moisture can be a strong control on carbon
cycling (Gatti et al., 2014), using SIF includes early stage ecosystem response to moisture availability in the VPRM in a way that traditional vegetation indices do not (Zhang et al., 2018a). Similar to Luus et al. (2017), our work adapts the VPRM formulation where the traditional MODIS-based satellite surface reflectance data is entirely replaced with SIF data derived from OCO-2 measurements (VPRM-SIF). In addition, our work further adapts the respiration term of the VPRM similarly to Winbourne et al. (2022) and Gourdji et al. (2022) where the amount of living biomass per pixel informs the autotrophic
respiration term. The adaptation of the VPRM respiration term addresses a tendency for the VPRM to systematically underestimate autotrophic respiration, at least in the extratropics (e.g., Dayalu et al., 2018, Gourdji et al., 2022). In the tropics, where respiration is a strong control on NBE, models that provide more accurate respiration parameterizations are needed. The biogenic carbon fluxes generated by SIF-based VPRM formulations provide the basis for this study's quantification and categorization of the spatial and temporal variations in the Amazon's carbon sink strength from 2010–2020.

In our study, we expand upon the latest research for the Amazon, leveraging observations where available, to construct, improve, and evaluate biosphere carbon fluxes. This is a critical first step to evaluating long-term anthropogenic impacts on and trends in the Amazon's carbon balance. We broadly define Amazonia's wet and dry seasons as six-month periods, with the wet season typically extending from December through May, and the dry season extending from June through November. We construct and evaluate multiple versions of the VPRM: the traditional version before and after modifying respiration
(VPRM_TRA, VPRM_TRG); and the SIF-based version before and after modifying respiration (VPRM_SIF, VPRM_SIFg). The VPRM $CO_2$ fluxes are generated for all hours from 2014 through September 2020. As an additional test, we compare all versions of the diagnostic VPRM to output from the process-based bottom-up Simple Biosphere 4 model (SiB4; Haynes et al., 2021), which to our knowledge is the only publicly available hourly partitioned biogenic carbon flux estimate for the Amazon that extends through at least 2018. The hourly VPRM fluxes also include two main events which, as suggested by Liu et al.
(2017) and Brando et al. (2020), are increasingly representative of the Amazon's future conditions and impacts on the global carbon balance: the strong El Niño in 2015–2016, and the severe dry-season fires in 2019/2020. In this paper, we additionally evaluate model performance during March 2016 by constraining with OCO-2 land nadir/land glint (LNLG) total columns and evaluating against $CO_2$ observations from aircraft vertical profiles. The March 2016 case study was selected for three main reasons: (1) it falls well within the Amazon wet season such that the biosphere dominates land-atmosphere $CO_2$ exchange such
that model-measurement mismatch can largely be attributed to the biosphere flux model alone rather than entangling the influence of fires, and (2) it falls at the tail end of the severe 2015/2016 El Niño and more strongly illustrates potential strengths



and weaknesses in how the various models capture the impacts of temperature and water stress on Amazonian land-atmosphere carbon exchange; and (3) there was distinct and separable fire activity which provides an opportunity to evaluate the capacity of the biogenic flux models to separate NEE from fire fluxes. Figure 1 displays our study domain along with key modeled and

observational data sets used in top-down constraints and evaluations.

## 2 Methods

### 2.1 Data Processing Tools

We used R v4.2.1(https://cran.r-project.org/) for all data processing and analysis. We used the Weather Research and Forecasting Model (WRF) version 3.8.1 to model the meteorology driving the Stochastic Time-Inverted Lagrangian Transport

(STILT) model. WRFv3.8.1 configuration is detailed in Table S1.

### 2.2 VPRM: Estimating Vegetation $CO_2$ Fluxes

VPRM $CO_2$ fluxes from vegetation are modelled as NEE, GPP, and $R_{eco}$ and reported as µmol $CO_2$ m$^{-2}$ s$^{-1}$ (hourly, 12 km resolution) with the convention of negative fluxes indicating uptake from the atmosphere and positive fluxes indicating release

to the atmosphere.

### 2.2.1 Model Overview

The traditional and SIF-based VPRM formulations—without (VPRM-TRA, VPRM-SIF) and with (VPRM-TRG, VPRM-SIFg) respiration term modifications—provide hourly fluxes of biogenic $CO_2$ NEE partitioned into GPP and $R_{eco}$. Equations

1(ab) and 2(ab) describe all VPRM formulations.

$$NEE_{VPRM\_TRA} = -\left(\lambda \times T_{scale} \times \frac{1}{1+\left(\frac{PAR}{PAR_0}\right)} \times PAR \times P_{scale} \times W_{scale} \times EVI\right) + (\alpha \times T + \beta) \qquad (1a)$$

$$NEE_{VPRM\_TRG} = -\left(\lambda \times T_{scale} \times \frac{1}{1+\left(\frac{PAR}{PAR_0}\right)} \times PAR \times P_{scale} \times W_{scale} \times EVI\right) + ((\alpha \times T + \beta) + (\gamma \times EVI)) \qquad (1b)$$

$$NEE_{VPRM\_SIF} = -\left(\lambda \times T_{scale} \times \frac{1}{1+\left(\frac{PAR}{PAR_0}\right)} \times PAR \times \frac{SIF}{\cos(SZA)}\right) + (\alpha \times T + \beta) \qquad (2a)$$

$$NEE_{VPRM\_SIFg} = -\left(\lambda \times T_{scale} \times \frac{1}{1+\left(\frac{PAR}{PAR_0}\right)} \times PAR \times \frac{SIF}{\cos(SZA)}\right) + \left((\alpha \times T + \beta) + \left(\gamma \times \frac{SIF}{\cos(SZA)}\right)\right) \qquad (2b)$$



The bolded parameters λ (ecosystem light use efficiency, units: μmol $CO_2$ m$^{-2}$ s$^{-1}$/μmol Photosynthetically Active Radiation (PAR) m$^{-2}$ s$^{-1}$); PAR$_0$ (ecosystem half-saturation value of PAR, units: μmol m$^{-2}$ s$^{-1}$); α (ecosystem respiration temperature dependence, units: μmol $CO_2$ m$^{-2}$ s$^{-1}$ °C$^{-1}$); β (μmol $CO_2$ m$^{-2}$ s$^{-1}$; flux-weighted mean size of the respiring carbon pools), and γ (μmol $CO_2$ m$^{-2}$ s$^{-1}$; sensitivity to water stress and above-ground biomass) are all optimized from non-linear least squares (NLS) fits to ground-based measurements (e.g., Mahadevan et al., 2008; Gourdji et al., 2022). Domain-wide hourly

fluxes are then derived for each land use category present. Our domain land categories are based on the International Geosphere-Biosphere Programme (IGBP) and obtained from the MODIS MCD12Q1 0.05° gridded product. Calibration and land classification details are provided in Sect. 2.2.2.

     In all formulations, hourly photosynthetically active radiation (PAR, units: μmol m$^{-2}$ s$^{-1}$; Eq. 3) drives the diurnal flux signal and T$_{scale}$ (unitless; see Eq. 4) provides ecosystem temperature sensitivity. When scaling fluxes to the entire domain,

ERA5 meteorology (https://registry.opendata.aws/ecmwf-era5/; Accessed 9 Aug 2023) at 31 km horizontal resolution provides the hourly downwelling shortwave radiation (SW) used to derive diurnal PAR; 2-meter surface air temperature estimates (T$_{air}$) are used to calculate T$_{scale}$.

$$PAR = SW / 0.505 \tag{3}$$

$$T_{scale} = \frac{(T_{air} - T_{min}) \times (T_{air} - T_{max})}{\left[ (T_{air} - T_{min}) \times (T_{air} - T_{max}) - (T_{air} - T_{opt})^2 \right]} , \tag{4}$$

     As shown in Eq. 4, T$_{scale}$ modifies the GPP term by weighting the differences between observed T$_{air}$ and the ecosystem-specific requirements for minimum (T$_{min}$), maximum (T$_{max}$), and optimal (T$_{opt}$) temperatures for photosynthesis. T$_{min}$, T$_{max}$, and T$_{opt}$ values are informed by literature values relevant to the tropics (Ma et al., 2017; Slot & Winter, 2017; Tan et al., 2017).

For all tropical ecosystems, T$_{min}$ is set to 4°C. Following work by Slot & Winter (2017) we broadly divide our ecosystem classes into tropical dry and tropical wet ecosystems. Tropical dry ecosystems (pixels in domain with IGBP categories of woody savannas, savannas, and grasslands) are assigned a T$_{max}$ of 41.8°C (SD=2.1°C) and T$_{opt}$ of 30.7°C (SD=1.1). Tropical wet ecosystems (pixels in domain with IGBP category of evergreen broadleaf) are assigned a T$_{max}$ of 40.1°C (SD=1.8°C) and T$_{opt}$ of 29.8°C (SD=0.9°C).

In the traditional formulations (Eqs. 1a and 1b) the unitless scalars P$_{scale}$ (phenology; Eqs. 4a-c), W$_{scale}$ (water stress; Eq. 5), and EVI (surface "greenness"; Eq. 6) modify GPP based on remotely sensed measurements of surface reflectance in the near infrared ($\rho_{nir}$), shortwave infrared ($\rho_{swir}$), blue ($\rho_{blue}$), and red ($\rho_{red}$) bands.

$$P_{scale,non-evergreen} = \frac{1 + \frac{\rho_{nir} - \rho_{swir}}{\rho_{nir} + \rho_{swir}}}{2} \tag{4a}$$

$$P_{scale,evergreen} = 1 \tag{4b}$$



$$P_{scale,water} = 0 \tag{4c}$$

$$W_{scale} = \frac{1+\frac{\rho_{nir}-\rho_{swir}}{\rho_{nir}+\rho_{swir}}}{1+max(\frac{\rho_{nir}-\rho_{swir}}{\rho_{nir}+\rho_{swir}})} \tag{5}$$

$$EVI = 2.5 \times \frac{\rho_{nir}-\rho_{red}}{\rho_{nir}+(6\times\rho_{red}-7.5\times\rho_{blue})+1} \tag{6}$$

The SIF-based VPRM formulations (Eqs. 2a and 2b) entirely replace the $P_{scale} \times T_{scale} \times EVI$ quantity with instantaneous spatially contiguous SIF derived from OCO-2 measurements (4-day, 0.05° resolution as in Zhang et al., 2018a). However, as shown in Eq. 7, raw SIF has a PAR dependence (Zhang et al., 2018b; Zhang et al., 2020) which requires normalizing by top-of-canopy PAR—approximated as the cosine of the solar zenith angle (SZA) at the local OCO-2 overpass time of 1330h—before the diurnal PAR signal can be meaningfully introduced as written in Eqs. 2a-b.


$$SIF = PAR \times fPAR_{chl} \times EF \tag{7}$$

In Eq. 7, $fPAR_{chl}$ is the fraction of PAR absorbed by chlorophyll and EF is the fluorescence efficiency. Daily SZA data for each calibration site was obtained from the NASA Horizons portal (https://ssd.jpl.nasa.gov/horizons/app.html; Accessed
15 Feb 2021). When scaling to our domain, we use the solarPos package in R to calculate the average SZA at each pixel at 1330 local overpass time averaged over each four-day interval of the contiguous SIF product.

Finally, two of our VPRM formulations (Eqs. 1b and 2b) also incorporate a γ (μmol $CO_2$ m$^{-2}$ s$^{-1}$) term adapting recent modifications that better represent R$_{eco}$ responses associated with biomass seasonality and/or water stress (Winbourne et al., 2022; Gourdji et al., 2022). In the tropics, Clark et al. (2013) note that temperature sensitivity of night-time plant respiration can have a significant control on aboveground biomass production. As shown in Eq. 7, Gourdji et al. (2022) apply a quadratic
temperature formulation to the traditional VPRM for R$_{eco}$ which allows for a nonlinear temperature response; furthermore, $EVI$ and $W_{scale}$ are explicitly incorporated to account for aboveground biomass seasonality and water stress:

$$R_{eco} = (\boldsymbol{\alpha_1 T_{air} + \beta}) + (\alpha_2 T_{air}^2) + (\boldsymbol{\gamma EVI}) + (\theta_1 W_{scale}) + (\theta_2 W_{scale} T_{air}) + (\theta_3 W_{scale} T_{air}^2) \tag{8}$$


However, given both the spatial and temporal limitations of our Amazon region calibration sites (see Sect. 2.1.2), applying the Gourdji et al. (2022) respiration parameterization risks overfitting Amazonian respiration to a small number of specific sites. Furthermore, Winbourne et al. (2022) found that the VPRM respiration term approached the traditional $\alpha \times T + \beta$ linear result during the growing season in a USA urban environment. In tropical environments, where growing season conditions last
year-round, we therefore represent R$_{eco}$ with the bolded terms in Eq. 8—that is, as a linear function of air temperature with an





additional γ term that uses *EVI* or *SIF/cos(SZA)* to incorporate the influence of aboveground biomass seasonality and water stress.

**2.2.2 Domain Land Use Classification, Model Calibration, and Model Evaluation**

Prior to generating vegetation carbon fluxes for the entire study domain, the VPRM model is calibrated using NLS to
ground-based flux data collected for each major vegetation class. In our Amazonian study domain, ground-based data availability is an ongoing challenge; our calibration data was therefore restricted to publicly available data from the AmeriFlux and FluxNet eddy flux networks. Data from the eight available Amazon-relevant eddy flux sites span limited IGBP vegetation classes and subsets of time from 2001–2015 (Table 1, Fig. 2). With the exception of the tropical rainforest site Br-K67 (discussed later in this section), the eddy flux data is insufficient for separately fitting wet and dry seasons. Therefore, we
perform single NLS fits to calibration data which result in constant VPRM parameters for each ecosystem type over the study period. While the calibration data set is not dense enough to span both the spatial and temporal extents of our study, we note that it does provide valuable and sufficient information to establish reasonable *a priori* vegetation $CO_2$ flux estimates that can be meaningfully constrained with OCO-2 $CO_2$ observations for two main reasons. First, approximately 95% of the land use in the study domain from 2010–2020 is encompassed by the four IGBP categories for which eddy flux data was available,
namely: evergreen broadleaf (50%); savannas (20%); grasslands (20%), and woody savannas (5%) (Fig. 2b). Three of the eight sites represent the evergreen broadleaf ecosystems which is the most prevalent broad class in the domain. While we recognize the IGBP "evergreen broadleaf" categorization is an oversimplification of heterogeneous Amazonian evergreen ecosystems, we note the value in tuning of the VPRM to three separate evergreen broadleaf sites. Second, while the available eddy flux data can be offset up to thirteen years prior to the study period of 2014–2020, the major drivers of hourly ecosystem flux
variations are provided by $T_{air}$, PAR, surface reflectance indices, and SIF. That is, while the static ecosystem parameters of λ, $PAR_0$, α, β, and γ would benefit from tuning to eddy flux data seasonally and/or over the entire study period to reflect concurrent ecosystem states most accurately, the dominant source of real time variation is largely captured by $T_{air}$, PAR, surface reflectance indices, and SIF (Dayalu et al., 2018).

As shown in Table 1, each broad ecosystem type is assigned a representative calibration and evaluation site. The exception
is site Br-K34, which is the only representative of primary interior tropical rainforest and there was insufficient data to divide the data into calibration and evaluation years. As the remaining tropical rainforest sites (Br-K67, Br-K83) are impacted by higher degrees of disturbance and edge effects, comparison with Br-K34 was not appropriate. We therefore manually differentiate two categories of evergreen broadleaf/IGBP category 2: (1) western Amazon IGBP category 2 is designated as primary interior tropical rainforest and its pixels are represented by NLS parameters obtained from site Br-K34; (2) remaining
IGBP category 2 primary tropical moist forest occurring in domain is represented by NLS parameters obtained from site Br-K67 (Fig. S1).

As noted earlier, the long record of Br-K67—spanning 2002–2011—enables assessing uncertainty associated with using annual fit VPRM parameters rather than seasonally fit. We therefore used the Br-K67 record to evaluate the impact of



parameter seasonality on model performance. Accounting for data gaps, a total of seven years of hourly Br-K67 observations
were randomly separated into wet and dry seasons subsets for VPRM calibration (70% from each of wet and dry season data)
and evaluation (30% from each of wet and dry season data). We then performed seasonal NLS fits for each VPRM version.
We used the seasonal and annual fits in an assessment of model bias for the tropical rainforest/evergreen broadleaf class relative
to observations.

**Table 1. Eddy flux sites used for VPRM calibration and/or evaluation.** *Br-K34 site represents interior relatively undisturbed evergreen
broadleaf classes; K34 data is unvalidated due to insufficient representative data. [++]Br-PDG was the only member of IGBP 9 class and
separate years were used for calibration (2003) and evaluation (2001–2002).

| Site | Coordinates | Data Years | Land Use (IGBP Category) | VPRM Role | Data Source |
|---|---|---|---|---|---|
| Br-BAN | -9.82N, -50.1E | 2003–2006 | Seasonally flooded transitional forest/savanna (8/Woody Savannas) | Evaluation (CST) | Restrepo-Coupe et al. (2021) |
| Br-CST | -7.97N, -38.4E | 2014–2015 | Tropical dry forest/wet season cattle grazing (8/Woody Savannas) | Calibration | Antonino (2022) |
| Br-FNS | -10.8N, -62.4E | 2001–2002 | Pasture (10/Grasslands) | Calibration | Restrepo-Coupe et al. (2021) |
| Br-K34* | -2.61N, -60.2E | 2001–2002 | Primary interior tropical rainforest (2/Evergreen Broadleaf) | Calibration | Restrepo-Coupe et al. (2021) |
| Br-K67 | -2.89N, -55.0E | 2002–2011 | Primary tropical moist forest (2/Evergreen Broadleaf) | Calibration | Saleska (2019) |
| Br-K77 | -3.01N, -54.5E | 2001–2005 | Pasture/agriculture (10/Grasslands) | Evaluation (FNS) | Restrepo-Coupe et al. (2021) |
| Br-K83 | -3.02N, -55.0E | 2001–2004 | Primary tropical moist forest, selective logging (2/Evergreen Broadleaf) | Evaluation (K67) | Goulden (2019) |
| Br-PDG[++] | -21.6N, -47.6E | 2001–2003 | Savanna (9/Savannas) | Calibration (PDG-2003); Evaluation (PDG-2001, 2002) | Restrepo-Coupe et al. (2021) |

## 2.3 SiB4

The SiB4 model version 4.2 (SiB4) integrates land cover, phenology, dynamic carbon allocation, cascading carbon pools
from live biomass to surface litter to soil organic matter, and NASA MERRA2 meteorology (Haynes et al., 2021). Unlike the
VPRM it is a mechanistic model that is independent of satellite indices and site-specific tuning parameters (Haynes et al.,
2019). We obtained SiB4 global hourly carbon fluxes partitioned into GPP and $R_{eco}$ at 0.5x0.5deg resolution from 2000–2018
from the Oak Ridge National Laboratory Distributed Active Archive Center (ORNL DAAC; Haynes et al., 2021). SiB4 fluxes
for each gridcell are disaggregated into one of 15 plant functional types (PFTs). We convert SiB4 fluxes per PFT to total fluxes
per grid cell using the provided area per grid cell. We then subset these grid cell total SiB4 fluxes over the VPRM domain
(Fig. 1) and regrid to the VPRM model resolution using bilinear interpolation. Additional details and technical description of
the SiB4 model v4.2 are provided in Haynes et al. (2020).



**2.4 Regional Inversion Methodology**

In this study we apply the CarbonTracker-Lagrange regional inverse modeling framework Hu et al. (2019), Rastogi et al. (2021), Rastogi et al. (2021b). It uses a Bayesian inversion methodology with a geophysically-based simulation of satellite retrievals of column $CO_2$ profiles (Sec. 2.4.1), and sensitivities of the satellite observations to upstream surface fluxes (i.e. "footprints") derived from WRF-STILT (Sec. 2.4.2). The March 2016 period was extracted from a three-month inversion, spanning February 2016 through April 2016, to account for edge effects. On average, wet season OCO-2 data coverage was

~15% less than dry season coverage.

We use OCO-2 total column $CO_2$ (XCO$_2$) retrievals to constrain VPRM $CO_2$ flux estimates in a geostatistical Bayesian framework, where we write the posterior flux solution as:

$$s = s_a + QH^T(R + HQH^T)^{-1}(X_{CO2}^{ret} - X_{CO2}^{sim}), \tag{9}$$


where $X_{CO2}^{ret}$ is the $n \times 1$ vector of retrieved XCO$_2$. $X_{CO2}^{sim}$ (Eq. 10) is the $n \times 1$ vector of simulated XCO$_2$. $H$ (Eq. 11) is the column weighted sensitivity of $X_{CO2}$ to surface fluxes upwind of the measurement, otherwise known as the column weighted footprint $[n \times m]$ or Jacobian. $s$ is the posterior flux estimate $[1 \times m]$. $R$ is the model-data mismatch covariance matrix $[n \times n]$, which ideally includes uncertainty due to geostatistical measurements, forward model, model representation, and boundary conditions. Given

the difficulty in estimating these uncertainties, in our application, R is estimated from the uncertainty in XCO$_2$ using the methodology of Peiro et al. (2022) with some modifications, discussed next. $Q$ is the prior flux error covariance matrix $[m \times m]$, and in our application is calculated using the standard deviation of the temporally averaged prior fluxes, a temporal correlation length of 7 days, and a spatial correlation length of 1000 km. $s_a$ is the prior flux estimate $[1 \times m]$ from the VPRM or SiB4 flux models. In this study, 2644 retrievals ($n$) are used over the February to April 2016 time frame and 1266 land grid cells are

optimized over the domain, hourly, for 2160 hours ($m$=2,734,560).

Our methodology follows that for comparing satellite retrievals to simulated $CO_2$ columns from the Atmospheric $CO_2$ Observations from Space (ACOS) retrieval algorithm v10 (O'Dell et al., 2012). We follow the convolution method as detailed in Rastogi et al. (2021) to calculate the simulated XCO$_2$:

$$X_{CO2}^{sim} = \sum_{i=1}^{N} w_i \left[ \alpha_i * \left( X_{CO2,i}^{bkg} + H_i(s_{bio} + s_{other}) \right) + (1 - \alpha_i) * X_{CO2,i}^{pri} \right], \tag{10}$$

where N is the number of vertical levels. $w_i$ is the pressure weighting function calculated as in Rastogi et al. (2021). $\alpha_i$ is the retrieval averaging kernel profile. $X_{CO2,i}^{bkg}$ is the $CO_2$ background profile from NOAA's CarbonTracker version CT2019B Jacobson et al. (2020). $s_{bio}$ is the biospheric flux estimate from VPRM or SiB4 models. $X_{CO2,i}^{pri}$ is the OCO-2 prior $CO_2$ profile.

$H_i$ is the sensitivity function or footprint on discrete levels from WRF-STILT (Sec. 2.5.2):



$$H = \sum_{i=1}^{N} w_i \alpha_i H_i \tag{11}$$

In this application, N is 17 levels. This accounts for 14 levels of WRF-STILT footprints, discussed in section 2.5.2, and 3
levels in the upper atmospheric column where the impact of surface fluxes are assumed to be negligible and the footprints are
set to zero. Thus in the upper atmosphere, the $X_{CO2}^{sim}$ calculation is dominated by the CarbonTracker background profile and
OCO-2 prior $CO_2$ profile.

For this study, we assimilate land nadir and glint OCO-2 $XCO_2$ retrievals that are averaged to 2s along track
(~13.5km) – rather than 10s averages – to better match the higher spatial resolution of our inversion domain (12 km) compared
to global model resolutions. We average the OCO-2 $XCO_2$ 2s data following methodology from Peiro et al. (2022), where a
10s average dataset was used. However, for the application of 2s data over land we estimate the uncertainty in 2s $XCO_2$ (R)
using a higher land correlation coefficient of 0.7, compared to 0.3 in Piero et al. (2022). In our application we also neglect
transport model errors, which are difficult to quantify but can be further explored in the future using transport ensembles.
Errors in WRF-STILT transport are discussed in Rastogi et al., (2021) in terms of calculating $X_{CO2}^{sim}$ for OCO-2 $XCO_2$ retrievals
in North America. They found that low partial column bias relative to independent vertical profile $CO_2$ data show that errors
in WRF-STILT transport contribute very minimally to bias in $X_{CO2}^{sim}$.

### 2.4.2 Transport Model Framework

We use WRFv3.8.1 model transport fields to drive the STILT model dispersion and compute footprints. A footprint
($H_i$), in units of mixing ratio per unit flux (e.g., ppm $\mu mol^{-1} m^{-2} s^{-1}$), quantitatively describes how the surface fluxes upwind of
a mixing ratio measurement location are influencing the measurement. For each OCO-2 $XCO_2$ retrieval location, footprints are
computed at 14 individual levels throughout the column. These 14 footprints are then convolved with OCO-2's pressure
weighting function and averaging kernel, and interpolated to STILT levels. This produces one column weighted footprint (H)
per retrieval, which quantifies how the OCO-2 $XCO_2$ retrieval is influenced by surface fluxes upwind of the column. The
column weighted footprint is not to be confused with a satellite footprint, which describes the area of earth reflecting the
satellite signal. The computational domains of the two nested WRF grids are shown in Figure 1a. Model configuration details
are provided in Table S2.

STILT computes footprints by modeling the dispersion of 500 particles, 10 days back in time from each measurement
location (in this case the x,y,z,t coordinate of an $XCO_2$ retrieval), and in 14 levels from 50 m to 14 km AGL. where each
receptor is located at a column measurement latitude and longitude and on one of 14 levels (50m to 14km AGL). Back
trajectories computed by STILT are affected by both resolved wind velocities and parameterized sub-grid scale turbulent
motions and convective fluxes. The footprint domain is outlined in Figure **Error! Reference source not found.**b, where



footprints are aggregated to 1 degree resolution at 1-hour intervals, compatible with the VPRM prior flux resolution used in the inversion.

## 2.5 Model Aircraft Vertical Profile Calculation

We calculate prior and posterior modelled vertical profiles for SiB4 and each VPRM formulation for our March 2016 optimization period at locations roughly corresponding to the vertical profiling sites RBA and ALF displayed in Fig.1a. As the available OCO-2 receptors are not identical in space and time to locations of RBA and ALF profiling sites, a direct comparison of simulated prior and posterior $CO_2$ vertical profiles derived from convolving multi-level OCO-2 receptor footprints and vegetation flux models is not possible. In addition, RBA and ALF vertical profiles are typically obtained once or twice a month

such that robust measured monthly averages for a single month are not available. To allow for direct comparison between modelled (prior and posterior) and measured vertical profiles, we construct pooled datasets that occasionally combine February and March 2016 measurements and/or modelled fields to develop a dataset that adequately represents a typical 2016 wet season month. Given the seasonal similarities across the Amazon in February and March, combining data across these months to create a representative "typical wet season month" is reasonable. Our method is detailed below.

We first construct a pooled dataset of "typical wet season 2016" measured vertical profiles at each of RBA and ALF sites. For RBA, we combine all measurements obtained between February and March 2016 (2016-02-08 at 1630UTC; 2016-02-27 at 1645UTC; and 2016-03-17 at 1700UTC), resulting in three measurements at each of 17 vertical levels between 300 and 4500 m asl. For ALF, we combine all measurements obtained between February and March 2016 (2016-02-23 at 1630UTC; 2016-02-29 at 1540UTC; 2016-03-13 at 1600UTC; and 2016-03-30 at 1540 UTC), resulting in four measurements at each of

12 vertical levels between 450 and 4500 m asl. Second, we assess the availability of OCO-2 footprints in the vicinity of RBA and ALF sites from February to March 2016. Our goal was to obtain footprints sufficiently close to each profiling site to be representative of the near-field influences on the site, but also have a large enough bounding box so that at least two OCO-2 receptors and their footprints were present for transport uncertainty calculations. Figure 2a displays the final selected 5x5 km bounding boxes around each of RBA and ALF and the representative OCO-2 receptors. RBA and ALF simulated vertical

profiles are then derived from the OCO-2 receptor footprints bounded in each box (Fig. 2a). Third, we use NOAA's web-based HYSPLIT model to assess land surface influences on each of the measured vertical profile dates and compare them with the land surface influences on each of the selected OCO-2 receptors. Figures S2-S3 show that the land surfaces influencing both the simulated and actual profiles at RBA and ALF are comparable with the airmass trajectories representing typical seasonal prevailing winds in the February/March 2016 time frame and annually from 2010–2018; the average air mass trajectories are

displayed in Fig. 2a. Fourth, we pool all vertically resolved CT2019 background concentrations associated with each footprint file's endpoints. Next, we use the March 2016 prior and posterior hourly fluxes to estimate a "typical wet season 2016 month of fluxes" and convolve them with each of the six OCO-2 receptors in the RBA and ALF bounding boxes to obtain a spread of enhancements or depletions relative to the CT2019 background $CO_2$ from 10 days prior. Finally, we linearly interpolate all components on WRF-STILT vertical grids to each of the RBA and ALF vertical grids; calculate a month's worth of hourly



model-observation residuals at each vertical level; and extract means, 25th, and 75th percentiles. We bootstrap CT2019 background concentrations and vertical profile measurements at each vertical level to dimensions that enable merging with the month of hourly $footprint \times flux$ fields. Ultimately, vertical profiles of modelled and measured residuals for the simulated month typical of the 2016 wet season incorporate uncertainties in transport, background, vertical profile measurements, and flux fields.

## 3 Results and Discussion

### 3.1 VPRM Calibration and Evaluation

    Site-specific values of tuning parameters are provided in Table S1. NLS calibration results are summarized in Table 2. Table 2 provides the root mean square error (RMSE) of predicted vs. measured NEE from the NLS fits at each calibration site, along with the interquartile range (IQR) of measured NEE which allows for normalizing RMSE across the varied ecosystems present in the domain. Generally, relative to the traditional VPRM formulations, the VPRM-SIF formulations provide improved fits (calibration) and predictions (evaluation) to hourly NEE observations. Of the VPRM formulations with SIF, the SIFg formulation is a slight improvement over the standard SIF formulation in net. An interesting feature of the respiration parameterization—relative to the typical non-tropical VPRM formulations—is the estimate of negative values for α in the respiration temperature dependency for the tropical dry forest Br-CST and interior primary rainforest Br-K34 sites. This is likely a response to the strength of temperature as a driver of NEE and respiration: strong decreases in respiration in certain tropical ecosystems have previously been observed (Clark et al., 2013; Gallup et al., 2021).

    We note that Br-K83—the Br-K67 evaluation site—is a selectively logged disturbed forest site (Figueira et al., 2008). We also note that site Br-K77 is a highly disturbed pasture/cropland site with a history of forest conversion to cropland in the 1990s followed by burning for rice cultivation in 2001. Br-K77 is therefore limited in its representation of steady-state cropland/grassland mosaic ecosystems: its poor performance as an evaluation site for the FNS cropland/grassland mosaic reflects its disturbed ecological history in the limited years for which we have eddy flux data. Specifically, as we have K77 eddy flux data only for 2001–2005, we are comparing an unstable ecology with FNS which began its trajectory to cropland as far back as 1977 (Almeida et al., 2018).

    Given the greater importance of respiration on the NEE (and NBE) signal in the tropics relative to the extra-tropics, we also examine differences in respiration parameterization across all VPRM formulations. We use night-time NEE as a proxy for respiration (i.e., absent confounding effects of daytime uptake) focusing on the hours from 20:00 LST to 03:00 LST. Overall, we find that annually and by season, the SIF-based VPRM formulations in general—and VPRM-SIFg in particular— have consistently better agreement with observed (night-time) respiration than the traditional formulations (Fig. 3, Table S3). On average, dry season respiration is better constrained across all model versions. In the wet season, while all models tend to underestimate respiration, VPRM-TRA displays the greatest bias. The underestimate in wet season night-time respiration also implies that the general underestimate of peak daytime wet-season drawdown in NEE occurs through underestimating GPP



rather than through over-estimating $R_{eco}$. Of the VPRM formulations, the SIF-based formulations have more instances of overestimating respiration, especially in savanna ecosystems (Table S3).

Diurnally, the SIF-based formulations consistently perform better than the traditional formulations when compared to observations annually and seasonally (Figs. S2–S3). Notably, for evergreen broadleaf classes—the dominant domain vegetation type—peak uptake tends to be underestimated in both seasons but especially in the wet season. We discuss this further in Sect. 3.1.1.

**Table 2. Root mean square errors and correlation coefficients ($R^2$) from NLS fits to calibration sites and associated evaluation site predictions.** For evaluation sites, the site used for calibration is provided in the Cal Site column. Calibration (model fitting) results with $R^2 > 0.5$ are in *regular italics.* Evaluation (model predictions) with $R^2$ values $>=0.5$ are in ***bold italics.*** VPRM model version shown as one of TRA, TRG, SIF, SIFg. *K34 data is unvalidated due to insufficient representative data. [+]K77 ecological history suggests limitations in its representation of steady-state grasslands. [++]PDG was the only member of IGBP 9 class and separate years were used for calibration (2003, $PDG_a$) and evaluation (2001–2002, $PDG_b$). The InterqIQR of measured NEE at each site is also provided to normalize RMSE and facilitate
comparison across the varied ecosystem types.

| Site | Cal Site | IGBP Ecosystem/Number | $NEE_{IQR}$ | TRA RMSE ($R^2$) | TRG RMSE ($R^2$) | SIF RMSE ($R^2$) | SIFg RMSE ($R^2$) |
|---|---|---|---|---|---|---|---|
| Br-BAN | BR-CST | Woody Sav/8 | 17.2 | 11.4 (0.29) | 11.2 (0.34) | 10.48 (0.42) | 10.39 (0.43) |
| Br-CST | -- | Woody Sav/8 | 4.95 | 3.88 (0.43) | 3.77 (0.45) | *3.60 (0.50)* | *3.36 (0.60)* |
| Br-FNS | -- | Grasslands/10 | 15.8 | 8.21 (0.44) | 8.18 (0.44) | *7.33 (0.60)* | *7.31 (0.60)* |
| Br-K34* | -- | Evergreen Bdlf/2 | 21.2 | ***8.30 (0.61)*** | ***8.20 (0.62)*** | ***8.55 (0.59)*** | ***8.28 (0.62)*** |
| Br-K67 | -- | Evergreen Bdlf/2 | 19.9 | ***6.05 (0.69)*** | ***5.58 (0.74)*** | ***5.67 (0.74)*** | ***5.30 (0.77)*** |
| Br-K77[+] | BR-FNS | Grasslands/10 | 7.81 | 6.84 (0.08) | 6.63 (0.13) | 6.58 (0.15) | 6.57 (0.16) |
| Br-K83 | BR-K67 | Evergreen Bdlf/2 | 19.4 | *7.59 (0.60)* | *7.28 (0.63)* | *7.00 (0.65)* | *6.79 (0.67)* |
| Br-PDG$_a$[++] | -- | Savannas/9 | 8.33 | ***3.89 (0.51)*** | ***3.87 (0.51)*** | ***3.93 (0.53)*** | ***3.91 (0.53)*** |
| Br-PDG$_b$[++] | BR-PDG$_a$[++] | Savannas/9 | 7.46 | ***3.98 (0.58)*** | ***4.05 (0.55)*** | ***4.01 (0.60)*** | ***4.08 (0.58)*** |

### 3.1.2 Impact of Seasonality on VPRM Performance: Br-K67

Wu et al. (2017) explore causes of GPP variability in Amazonian rainforests, focusing on the relative importance of light versus water limitations at the Br-K67 site, using data spanning 2002–2011. Wu et al. (2017) estimate monthly and constant light use efficiency (LUE; $CO_2$ uptake per unit of PAR) directly from the ratio of observationally derived GPP to PAR under reference (i.e., "non-stressed") environmental conditions. Their resulting reference monthly LUE ranges from 0.02 to 0.03 $\mu$mol $CO_2$ ($\mu$mol photons)$^{-1}$ over the seven years of Br-K67 eddy flux data in contrast to a constant LUE of ~0.02 $\mu$mol $CO_2$
($\mu$mol photons)$^{-1}$. Wu et al. (2017) show seasonality in monthly LUE, with peak dry season values lower than peak wet season by as much as 33%. In this section, we present our results examining (1) seasonal variations in the LUE (i.e, the $\lambda$ term in VPRM) term across model versions and compared to literature values; and (2) impacts of seasonality in VPRM fitting parameters on modelled relative to observed NEE both in aggregate and diurnally.





Figure 4a displays the monthly average LUE obtained from VPRM NLS fitting to all available Br-K67 data, with LUE
estimated by Wu et al. (2017) included for reference. We note that all versions of the VPRM indicate a seasonal cycle in LUE
whose timing is similar to Wu et al. (2017). However, while the traditional VPRM formulations estimate a peak dry season
LUE that is 66% lower than the peak wet season LUE, the difference in SIF-based formulations is 39–46% lower in the dry
season. The traditional VPRM versions estimate monthly LUE that is frequently an order of magnitude higher than those
estimated by the SIF-based versions and in Wu et al. (2017). In contrast, both SIF-based VPRM formulations estimate LUE
that is generally higher but comparable to Wu et al. (2017); differences are ascribed to the LUE estimation methodology.
Overall, the LUE from SIF-based VPRM formulations have significantly better agreement with Wu et al. (2017); given the
direct relation between LUE and GPP, our results provide higher confidence in SIF-based GPP estimates over the traditional
VPRM GPP estimates. Seasonal variations in diurnally averaged modeled PAR are shown in Fig. 4b-c.

Given the seasonality in tropical forest LUE, and their dominance in domain land cover type, we next test our assumption
that (1) the VPRM environmental drivers (i.e., SIF, Tscale, Pscale, EVI, LSWI, $T_{air}$) explain more of the real-time variation in
carbon flux than the tuning parameters, and (2) constant annual-based tuning parameters can reasonably be used across seasons
and years. Wu et al. (2017) have previously shown that while environmental drivers indeed explain the most variability at
shorter (hourly) timescales, they tend to explain progressively less at longer timescales (diurnal, monthly, annual) when the
influence from intrinsic ecosystem variables begin to aggregate. However, Wu et al. (2017) use an LUE-based photosynthesis
model that is adopted from the VPRM-TRA formulation as in Mahadevan et al. (2008).

**Table 3. Impact of calibration parameter seasonality on NEE predictability.** Calibration parameters used in evaluation are obtained
from the annual set where NLS fitting was conducted for entire Br-K67 data set ($NEE_{ann}$), the wet season where calibration was conducted
for a random 70% subset of Br-K67 wet season data ($NEE_{wet}$), and the dry season where calibration was conducted for a random 70% subset
of Br-K67 dry season data ($NEE_{dry}$). Mean Bias is reported for each of wet and dry seasons relative to hourly NEE observations in the 30%
seasonal evaluation subsets. % Error (model relative to observations) and $R^2$ values are provided in parentheses.

| Model | Calibration Parameters | Mean Bias, µmol CO₂ m⁻² s⁻¹ (% error, *R²*) | |
|---|---|---|---|
| | | **$NEE_{wet,VPRM}$ vs. $NEE_{wet,Obs}$** | **$NEE_{dry,VPRM}$ vs. $NEE_{dry,Obs}$** |
| **VPRM-TRA** | **$NEE_{ann}$** | 1.92 (36.4, *0.72*) | -1.21 (18.9, *0.74*) |
| | **$NEE_{wet}$** | 0.041 (0.788, *0.72*) | -4.54 (70.9, *0.74*) |
| | **$NEE_{dry}$** | 2.81 (53.4, *0.72*) | 0.015 (0.230, *0.74*) |
| **VPRM-TRG** | **$NEE_{ann}$** | 0.402 (7.64, *0.77*) | -0.239 (3.73, *0.76*) |
| | **$NEE_{wet}$** | 0.020 (0.373, *0.77*) | -1.48 (23.0, *0.76*) |
| | **$NEE_{dry}$** | 0.456 (8.67, *0.77*) | -0.007 (0.103, *0.77*) |
| **VPRM-SIF** | **$NEE_{ann}$** | 0.451 (8.57, *0.71*) | -0.326 (5.09, *0.77*) |
| | **$NEE_{wet}$** | 0.064 (1.22, *0.71*) | -1.15 (17.9, *0.77*) |
| | **$NEE_{dry}$** | 0.591 (11.2, *0.77*) | -0.021 (0.324, *0.77*) |




| | | | |
|---|---|---|---|
| **VPRM-SIFg** | **NEE$_{ann}$** | 0.362 (6.88, *0.78*) | -0.173 (2.71, *0.78*) |
| | **NEE$_{wet}$** | 0.146 (2.79, *0.78*) | -0.592 (9.24, *0.78*) |
| | **NEE$_{dry}$** | 0.469 (8.92, *0.78*) | -0.00 (0.013, *0.78*) |

Table 3 summarizes the results from a test where the wet and dry season Br-K67 evaluation subsets (described in Sect. 2.2.2) were used to predict seasonal NEE from annual- and seasonal-fit calibration parameters (Table S1). Overall, while all models generally underestimated net uptake in the wet season and underestimated net release in the dry season, the VPRM-SIFg formulation performed consistently better across all calibration parameters with hourly errors ranging from 0-10%. As expected, the optimal model-observation fit across all model versions occurred when season-specific parameters were used (% error < 3). However, when seasonal NEE was predicted using calibration parameters tuned to the opposite season, VPRM-TRA performed the worst (% error: 53–71), followed by VPRM-TRG (% error: 8.7–23), VPRM-SIF (% error: 11–18) and VPRM-SIFg (% error: 8.9–9.2). Finally, errors in seasonal NEE predictions from annual parameters smoothed the impacts of lower cross-season predictability; however, even with the cross-seasonal smoothing, the VPRM-TRA performed relatively poorly (% error: 19–36%), particularly when compared to VPRM-SIFg (% error: 2.7–6.9).

Figure 5 displays diurnal patterns in seasonal predictability. While the diurnal averaging by season smooths the hourly model-observation mismatch displayed in Table 3, the diurnal breakdown suggests improved respiration parameterization in the SIF-based VPRM formulations.

While Br-K67 seasonal predictions using annually fit parameters smoothed the impacts of lower cross-season predictability, the cross-season results (Table 3, Fig. 5) combined with the order-of-magnitude higher LUEs (Fig. 4a), indicate the traditional VPRM formulations are driven by environmental variables that inadequately represent the dominant land cover type in the study domain. The exceptionally poor VPRM-TRA results also brings into question the reliability of the VPRM-TRG formulation given that they both incorporate the same environmental drivers. We also note that across all model versions, model-observation mismatch was highest in the wet season which is intuitive given the added complexity of cloud cover and associated impacts on uncertainties in remotely sensed environmental drivers and light use efficiency. The results from the seasonality case study suggest that the SIF-based formulations in general—and the VPRM-SIFg formulation in particular—are better equipped at estimating biogenic carbon fluxes at both short and longer timescales. We explore these results further via the wet season 2016 optimization case study in Sect. 3.2.

Of note, while there can be conflicting evidence on light-limitations and seasonal carbon uptake in the Amazon, several studies provide evidence for enhanced dry-season carbon uptake when cloud cover is reduced and sunlight is more readily available (Huete et al., 2006; Saleska et al., 2016; Doughty et al., 2019). But while the GPP constituent of NEE can be higher in the dry season, wet season aircraft observations tend to show higher uptake in net (Gatti et al., 2010, 2014, 2021).

### 3.2 Wet Season 2016 Case Study: Biogenic Model Evaluation and Estimates of Fire Influence



We evaluate the VPRM performance during March 2016, corresponding to the middle of the wet season immediately following the peak of the 2015/2016 severe El Niño, and compare with SiB4 performance. We select the March 2016 wet season, as it is a valuable natural experiment that demonstrates the relative importance of biogenic fluxes and fire fluxes. The

severe El Niño corresponded to one of the hottest periods recorded over Amazonia in the past century; as shown in Fig. 6a, the 2016 wet season was characterized by a dipole of extreme drought in the eastern Amazon and unusual wetting in the western Amazon (Silva Junior et al., 2019). The anomalously hot and dry conditions in portions of the Amazon were more conducive to drought-related fire activity; however, Silva Junior et al. (2019) found the peak in Amazon area affected by fires during the first quarter of 2016 was approximately 2% with fire anomalies concentrated in the northern state of Roraima.

Integrated over February and March 2016, Aqua satellite fire detects were concentrated at the northern and southeastern boundaries of the Amazon Basin (TerraBrasilis, 2024). The emissions strength of these fires (as fire radiative power) was not readily available, and could not be used to weight their relative importance. Dominance of fire emissions relative to biosphere fluxes varied by location; we use this feature in combination with drought index information and measured aircraft vertical profiles to evaluate biogenic flux models and estimate the relative contribution of the biogenic activity and fires to sub-regional

$CO_2$ signal.

### 3.2.1 Optimization Results for Biogenic Flux Models

Figure 6 displays 1°×1° prior (Fig. 6b) and posterior (Fig. 6c) fluxes, and their differences (Fig. 6d), as μmol $CO_2$ m$^{-2}$ s$^{-1}$ averaged over the month of March 2016. The optimization suggests that all prior models in wet season 2016 tended to

underestimate carbon uptake in the western Amazon; the region of underestimated uptake corresponds to the anomalously wet region identified through the Self-calibrated Palmer Drought Severity Index (ScPDSI) by Jimenez-Muñoz et al. (2016) (Fig. 6a). Relative to OCO-2 total column observations, both traditional VPRM formulations tend to underestimate uptake throughout the Amazon and strongly underestimate uptake in the region corresponding to the wetter than normal conditions. However, incorporating SIF observations into the VPRM formulations enhances the model's ability to capture uptake in

anomalously dry conditions. We note the demonstrated drought sensitivity of SIF, providing confidence in its ability to quantify impacts of water availability on carbon uptake (e.g. Mohammadi et al., 2022).

Overall, our optimization results during the differentially drought-impacted 2016 wet season suggest that SIF-based VPRM formulations are better able to capture carbon uptake during varying tropical light and moisture regimes than the traditional surface-reflectance based VPRM formulations. We also find that the posterior solutions of the diagnostic SIF-based

VPRM formulations—and the SIFg formulation in particular—converge with that of the bottom-up process-based SiB4 model (Fig. 6d). We further note that the SIF-based VPRM formulations provide significantly higher spatial heterogeneity relative to the SiB4 model, suggesting improved parameterization of the complexity of Amazonian ecosystems.

### 3.2.2 Comparison of Prior and Posterior Models at Aircraft Vertical Profiling Locations





We evaluate the prior and posterior performance of the VPRM and SiB4 models at the RBA and ALF vertical profiling sites, focusing on altitudes where upwind biogenic fluxes dominated the $CO_2$ signal. After qualitatively accounting for likely presence of fire emissions, we examine the impact of biogenic model optimization through simulating vertical profiles, diurnal cycles, and total monthly fluxes. Methods to calculate vertical profiles and total monthly flux are described in Sects.2.6 and 2.7.

Overall, we find the SIF-based VPRM formulations and the SiB4 simulate vertical profiles with significantly better model-observation agreement than the traditional VPRM formulations. We discuss the details of the analysis below, noting that the RBA profiling location alone provided observations at altitudes where the biosphere is dominant. Fire activity in the region upwind of ALF was determined to be substantial enough at all altitudes such that biogenic model evaluation—i.e., evaluation absent other significant and confounding sources of $CO_2$—would not be valid. Figures 7 and 8 display modelled and measured

vertical profiles and their 25[th] and 75[th] percentiles for a typical 2016 wet season month at RBA and ALF, where "typical" is defined using data across February and March 2016 (Sect. 2.6). At RBA, this period corresponds to surface influence from across the northeast and central Amazon Basin which is a strong modifier of the background $CO_2$ signal through evergreen broadleaf uptake (Figs. 2a, S4, S6). At ALF, however, the Amazon Basin has significantly less influence on the air masses; savanna and woody savanna ecosystems outside the Basin dominate the surface influences on the advected background air

(Figs. 2a, Figs. S5, S7). As indicated by the IQR of NEE in Table 2, savannas and woody savannas are weaker modifiers of the $CO_2$ signal relative to Amazon Basin evergreen broadleaf classes. In addition, during the first quarter (Q1; January–March) of the calendar year, the typical RBA influence region (Fig. S4) is ~14% deforested, unlike the ALF Q1 influence region (Fig. S5) which is ~23% deforested during the same period (Gatti et al., 2021). The footprints from the four OCO-2 receptors used to simulate ALF vertical profiles, and the back trajectories from measured profiling locations, indicate highest ALF influence

from the most heavily deforested regions (Fig. 2a, Figs. S4-S7). As previously noted, numerous upwind fire locations potentially impact ALF at all altitudes such that the biosphere models alone cannot sufficiently approximate the observed concentrations at any altitude. Table 4 summarizes these results as differences between measured $CO_2$ vertical profiles and the modelled background. At altitudes <2000 masl, ALF Q1 upwind surfaces result in lower background $CO_2$ signal modification relative to RBA Q1 upwind surfaces, due to a combination of lower ecosystem carbon flux modification and higher fire

emissions. At altitudes >2000 masl, the differences between ALF and RBA measurements are not significant, suggesting similar background influences above that altitude. At altitudes from ~900–1000 masl, $CO/CO_2$ ratios from Gatti et al. (2021) indicate fire plume influence dominating the signal at RBA and ALF.

      The vertically resolved footprints in Figs. 7 and 8 and Table 4 suggest four distinct upwind $CO_2$ regimes influencing the RBA and ALF measured profiles: (1) biosphere flux dominance below 900 masl; (2) fire emission dominance between 900–

1000 masl (3) dilution of fire emissions by background air between 1000–2000 masl; and (4) background tropical marine Atlantic air dominance above 2000 masl. We note Gatti et al. (2021) surface influence regions are based on integrated back trajectories ranging from 300–3500 masl; while their upper threshold of 3500 masl defines tropical marine Atlantic background



air, their results were robust across background air designations from >1300 masl (typical regional planetary boundary layer height) to 3500 masl.

At RBA where the Amazon Basin biosphere dominates the signal more than at ALF, the optimization reduces the model-observation mismatch for all flux models, particularly below 1000 masl. Generally, the traditional VPRM formulations are significantly more impacted by the optimization than the SIF-based VPRM and SiB4 models. The magnitude at which VPRM_TRA and VPRM_TRG underestimate $CO_2$ uptake across the RBA area of influence results in a model-observation *a priori* surface residual of over 10 ppm and a posterior surface residual of approximately 5 ppm (Fig. 7a-b). In contrast,

VPRM_SIF, VPRM_SIFg, and SiB4 models are all similarly impacted by the optimization with *a priori* model-observation surface residual of approximately 5 ppm, and a posterior model-observation surface residual of <2.5 ppm. At ALF, where the February-March 2016 upwind air masses tend to bypass a majority of the Amazon Basin and are potentially significantly influenced by fire activity at all altitudes, all prior and posterior flux models perform similarly with a typical model-observation residual of -3 to 3 ppm throughout the vertical column (Fig. 8a–e).


**Table 4. Difference between average measured $CO_2$ profiles and CT2019 background $CO_2$.** Differences are based on averages pooled across February and March 2016. *1000 masl is approximate location of the peak of a fire plume.

|  | $CO_{2,OBS} - CO_{2,BG}$ (95% CI), ppm | | | |
|---|---|---|---|---|
|  | **<900 masl** | **~900–1000 masl\*** | **1200–<2000 masl** | **>2000 masl** |
| **RBA** | -3.2 (-3.5, -2.9) | -0.36 (-0.80, 0.084) | -0.46 (-0.72, -0.17) | -1.1 (-1.3, -0.97) |
| **ALF** | -1.7 (-2.0, -1.3) | 2.5 (2.0, 3.0) | -0.97 (-1.4, -0.56) | -0.99 (-1.2, -0.82) |
| ***ALF - RBA*** | *1.6 (1.1, 2.1)* | *2.8 (2.2, 3.5)* | *-0.52 (-1.0, -0.03)* | *-0.15 (-0.39, 0.10)* |

### 3.3 Summary of VPRM_SIFg Prior Model Decadal Performance

    Based on comparison with eddy flux data and results from the March 2016 optimization case study, the VPRM_SIFg

biogenic flux model shows promise in its ability to capture Amazonian carbon fluxes across multiple timescale and moisture regimes, suggesting its suitability for larger studies evaluating interannual and seasonal carbon trends in both the fire and biogenic components of the region's NBE. In this section, we summarize the decadal performance of the VPRM_SIFg prior relative to VPRM_TRG, VPRM_TRA and SiB4 priors. We calculate average seasonal fluxes of the VPRM_SIFg prior (as $\mu mol\ CO_2\ m^{-2}\ s^{-2}$) from 2010–2020 (2010–2018 for SiB4) over Amazonia and compare with the corresponding average fluxes

from the VPRM_TRA, VPRM_TRG, and SiB4 priors (Fig. 9).

### 3.3.1 Comparison with Traditional VPRM Fluxes

    Basin-wide, and across both seasons and all years, VPRM_SIFg estimates more photosynthetic uptake than VPRM_TRA and VPRM_TRG in an overall pattern consistent with model validation and wet season 2016 optimization results. Across both

seasons, VPRM-SIFg estimates more respiration in the Br-K34 calibration region, and less respiration across the rest of the basin when compared to VPRM-TRA and VPRM-TRG.





Outside of the Basin, VPRM-SIFg universally estimates higher photosynthetic uptake and respiration release than the traditional VPRM models.

### 3.3.2 Comparison with SiB4 fluxes

When compared to SiB4, VPRM-SIFg estimates less dry season uptake and respiration in the western Amazon Basin corresponding to the regions calibrated by Br-K34. In the wet season, VPRM-SIFg generally estimates less uptake and respiration than SiB4.

Outside of the Basin, VPRM-SIFg generally estimates more photosynthesis and respiration in the dry season. In the wet season, higher photosynthesis and respiration relative to SiB4 is localized to woody savanna/savanna ecosystems (Br-CST, Br-BAN).

### 4 Conclusions

Compounded by the impacts of global climate change, the Amazon is experiencing unprecedented ecological disturbance through fires, deforestation, drought, and forest fragmentation. Multiple recent studies describe how the increasingly degraded and disturbed Amazon has a lowered carbon sink capacity with significant impacts to regional and global carbon budgets. However, reliable biogenic flux models that can capture fluxes from hourly to annual timescales are needed to quantify trends in carbon sink strength; ecological health and recovery from disturbance; and estimates of emissions from increasingly prevalent fires.

Assimilating observational $CO_2$ data from eddy flux sites, OCO-2 columns, and aircraft vertical profiles, we demonstrate construct and evaluate the ability of four versions of the VPRM diagnostic light use efficiency model to capture biogenic carbon fluxes from hourly to seasonal scales. Of the VPRM versions evaluated, the respiration-modified VPRM_SIFg exhibits the least bias when compared to eddy flux observations, including observationally derived respiration estimates. In the tropics—where respiration exerts stronger controls on NBE than the extratropics—our work demonstrates the superior performance of the respiration-modified VPRM_SIFg diagnostic light use efficiency model relative to all other VPRM versions. In addition, optimization results for March 2016—the middle of the wet season corresponding to the tail of the 2015–2016 severe El Niño—demonstrate (1) significant underestimate of net uptake by the traditional VPRM formulations relative to observations and VPRM-SIFg; (2) relative to the SiB4 model, VPRM_SIFg model describes more spatial heterogeneity in carbon exchange throughout the Amazon; and (3) convergence of flux estimates from two distinct methodologies – namely, the diagnostic VPRM-SIFg and the process-based bottom-up SiB4 biogenic flux model. While the convergence of NEE in the distinct VPRM-SIFg and SiB4 flux estimation methodologies lends confidence to both models, differences in NEE partitioning (as GPP and $R_{eco}$) warrant further exploration.

We also find that, despite the paucity of wet-season OCO-2 observations used in top-down constraints, modelled vertical profiles optimized with OCO-2 measurements compare significantly better with aircraft measurements than their unoptimized



counterparts. We hypothesize that, given the drier than normal conditions experienced by the region influencing RBA during
this time period, the sounding density during this wet season was higher than normal. Future work focusing on a range of wet
season conditions will shed light on this possible anomaly.

The promising performance of the both the prior and posterior VPRM_SIFg model in the wet season 2016 case study
provides confidence in its ability to capture interannual and seasonal trends in the biogenic carbon component of the Amazon's
net biome exchange. We note that VPRM-SIFg would benefit from additional calibration sites, via inclusion of eddy flux data
from more recent time periods and/or through greater representation of interior primary moist evergreen broadleaf classes.
Currently, Br-K34 is the only site representing the interior Amazon; additional eddy flux data for calibration and/or validation
such as from Amazon Tall Tower Observatory (ATTO) would be beneficial. Future work will use the optimized VPRM-SIFg
across both wet and dry seasons to evaluate inter-annual and seasonal trends, including assessing the changing role of fires in
the net biome exchange of Amazonian carbon fluxes, and the changes in the pace of post-fire ecosystem recovery.


## Code availability

VPRM and inversion code are available upon request from the authors.

## Data availability

$CO_2$ vertical profiling data are available from https://doi.org/10.1594/PANGAEA.926834. VPRM fluxes, STILT footprints,
and CT2019 background data for the March 2016 case study are available from https://doi.org/10.7910/DVN/PJ1EVC.
Additional data available from authors upon request.

## Supplement

The supplement related to this article is available online.

## Author contributions

AD wrote the paper with input from all co-authors. AD constructed the VPRM. MM generated the WRF-STILT fields and
performed the inversion with assistance from BR and JBM. LVG provided access to vertical profiling data sets and shapefiles.

## Competing Interests

The contact author has declared that none of the authors has any competing interests.



**Disclaimer**

**Acknowledgements**

We thank Yao Zhang for assistance with the CSIF data set; Juan-Carlos Jiminez-Muñoz for providing the Amazon scPDSI data for 2016; and Sharon Gourdji, Matthew Alvarado, Rebecca Adams-Selin, and Thomas Nehrkorn for helpful discussion. AmeriFlux data were made available through the data portal (https://ameriflux.lbl.gov) and processing maintained by the AmeriFlux Management Project, supported by the U.S. Department of Energy Office of Science, Office of Biological and

Environmental Research, under contract number DE-AC02-05CH11231.

**Financial Support**

This research has been supported by the U.S. NSF Atmospheric Chemistry Program Grant No. 2026410; NASA OCO Grant No. 80NSSC24K0753; and NASA CMS Grant No. 80NSSC18K0171.



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



870

**Figure 1. Study spatial domain with OCO-2 and aircraft observation locations**. (a) WRF domain. The black diamonds indicate the semimonthly aircraft vertical profiling areas available for evaluation over the study period. Bounding boxes represent the VPRM (green, 12 km native resolution) and WRF meteorology domain nests (black: d01, 36km resolution; blue: d02, 12km resolution). The orange dashed line approximates the Arc of Deforestation. OCO-2 column measurements (nadir and glint land soundings, March 2016), thinned to 2s along-track, are provided as black points. (b) Vertically integrated footprints averaged for March 2016 optimization case study using 14 levels of WRF-STILT footprints within WRF d02 domain.





**Figure 2. IGBP land use categories in domain and influences on measurements.** (a) IGBP categories mapped to study domain, with eddy flux calibration/validation sites and OCO-2 receptors overlaid. Vertical profiling sites RBA and ALF, their representative OCO-2 receptors, and their upwing influences are highlighted. (b) Land-use percentages in domain.



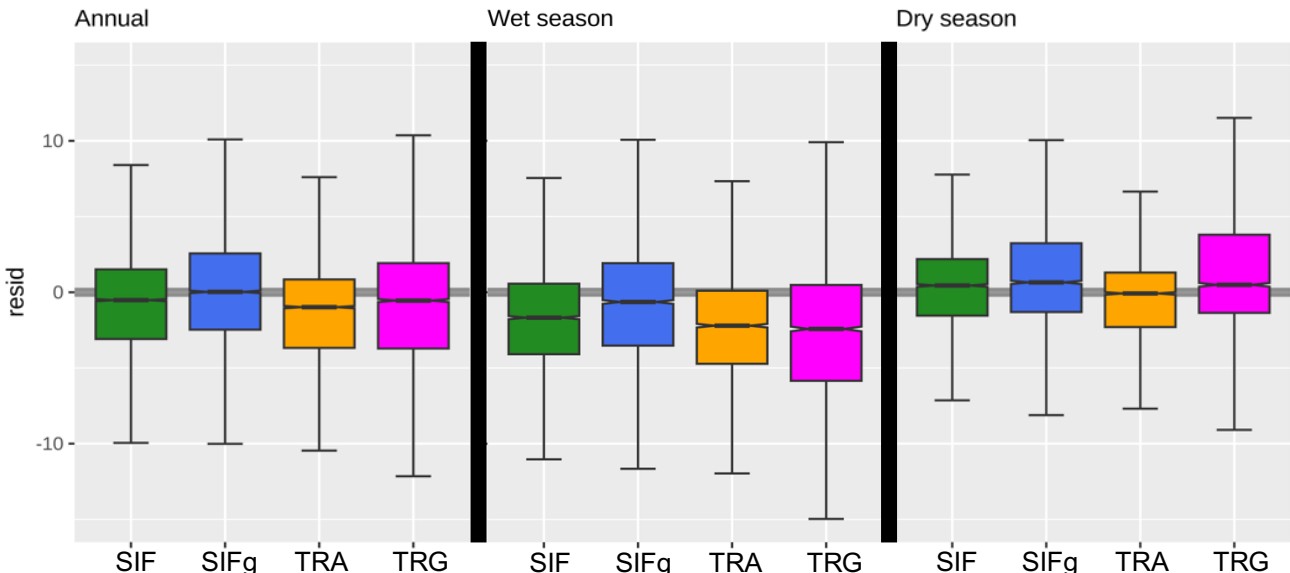

**Figure 3. VPRM Model-Observation (Night-time) Respiration residuals, annually and by season (μmol CO2 m$^{-2}$ s$^{-1}$).** Observed respiration is set to nighttime NEE where GPP = 0.



905



**Figure 4. Light Use Efficiency and ERA5 modeled PAR at Br-K67.** (a) Monthly and constant LUE estimated by VPRM and Wu et al. (2017); (b) modelled PAR.



**Figure 5. Seasonal fits to Br-K67 and diurnal model performance.** Left panels: wet season; right panels: dry season.





910



**Figure 6. Correlation of March 2016 regional drought patterns with differences between average monthly prior and posterior fluxes.**
(a) Self calibrated Palmer Drought Severity Index (scPDSI) from January through March 2016 exhibiting anomalously wet conditions in the western Amazon and anomalously dry conditions in eastern Amazon; (b-d) spatial patterns of average March 2016 fluxes for the (b) prior models, (c) posterior models, and (d) their differences. scPDSI data from Jiminez-Muñoz et al. (2016).





**Figure 7. Modelled and measured vertical profiles for a typical month during 2016 wet season at RBA aircraft site.** (a-e) Prior and posterior residuals of modelled vertical profiles relative to measured profiles; (f) measured $CO_2$. Shaded regions are 25th and 75th percentiles of bootstrapped distributions.





**Figure 8. Modelled and measured vertical profiles for a typical month during 2016 wet season at ALF aircraft site.** (a-e) Prior and posterior residuals of modelled vertical profiles relative to measured profiles; (f) measured $CO_2$. Shaded regions are 25th and 75th percentiles of bootstrapped distributions.







**Figure 9. Average GPP and R$_{eco}$ flux differences between VPRM-SIFg relative to traditional VPRM formulations and SiB4. Averaging period is 2010–2020 (comparison with VPRM formulations) and 2010-2018 (comparison with SiB4).** Top Panels: Dry season differences; Bottom Panels: Wet season differences. Blue (red) values indicate instances where SIFg estimates more (less) uptake and release than the comparison model.