# Peer review of "Constraining 2010-2020 Amazonian carbon flux estimates with satellite solar-induced fluorescence (SIF)"

_EGUsphere, 2024_

## Author Comment (AC1)

**Response to RC1**

**Review of the manuscript titled "Constraining 2010-2020 Amazonian carbon flux estimates with satellite solar-induced fluorescence (SIF)" by Dayalu et al.**

**I find the manuscript quite long, but given the data and methods used, the length is justified. The manuscript is well-written, with extensive and intensive data usage (Flux sites, Aircraft profiles, SIF data, XCO2 data from satellites and different NEE simulations). The analysis performed is robust with clear results. The discussion is quite thorough as well, but more detailed discussion on the limitations of the calibration process, especially given the spatial and temporal variability in the Amazon would be desirable.**
**I recommend minor revisions and have following questions/clarifications?**
We thank Reviewer 1 for the comments. The recommended changes have been made and have strengthened the paper. We have also provided clarifications as requested with some being included in the text as additional text or Figures as necessary. Where changes in the paper have been made, comments have been included as "RC1.#".

**RC1.1. Is there a reason why the authors did not extend the analysis from 2001-2020 (instead of 2010-2020)? I see that the CSIF is available since 2001, and the EC flux datasets are also available during the 2001-2010 period.**

We initially kept the analysis to the 2010-2020 period to reflect the availability of OCO-2 data for the top-down constraint (OCO-2 data availability began only in 2014). The overall goal was to constrain with OCO-2 back to 2014, and extrapolate the model-obs mismatch derived from inversion results by a few more years back to 2010. However, in-progress follow-on work is using additional data sets not previously available and indeed extending the VPRM back to the 2000s. The purpose of this initial work is to justify future transition to VPRM-SIF for the region.

**RC1.2. Equation 8: Although the authors clarify why they stick with the bold terms of the eq. 8 for the analysis, it would be great if there is some information about how the other terms of the equation affects the final NEE estimations.**

As found in Gourdji et al. (2021) – Figure 5 in that paper – the impact of the higher order terms is to increase the correlation coefficient $R^2$ with respect to night-time NEE (ie., night-time NEE is a proxy for respiration) measured at a given eddy flux site. It is expected that, for the Amazon eddy flux calibration sites, increasing the number of fitting terms will increase the $R^2$ but potentially by overweighting conditions specific to a given calibration site. Given the already sparse eddy flux representation for the vast and heterogeneous Amazon, calibrating multiple respiration parameters using a single site would be inappropriate. We had noted this in the text: namely, that absent a large network of eddy flux sites a quadratic formulation : "*However, given both the spatial and temporal limitations of our Amazon region calibration sites (see Sect. 2.1.2), applying the Gourdji et*

*al. (2022) respiration parameterization risks overfitting Amazonian respiration to a small number of specific sites."*

**RC1.3. Figures 1 & 6. Please get rid of the rainbow color palette (the rainbow colour map is not perceptually uniform) and consider replacing it with the color palette of Figure 9.**

Done. We have changed Figure 1b to Viridis; Figure 6bc to Inferno; and Figure 6d to a red/blue uniform palette suitable for displaying difference plots. Figures are reproduced below.
**Figure 1b:**

[Figure]

**Figure 6bcd:**

[Figure]

**RC1.4. When comparing the fluxes with SiB4 (section 3.3.2), it would be great if authors clarify whether these differences are due to the different model structures or input data, or whether they reflect genuine differences in carbon dynamics that should be explored further.**

We have now explicitly noted this at the end of Section 3.3.2: *"Determining the extent to which the differences between the two models reflect real carbon dynamics requires a multi-year optimization, including separately optimizing GPP and $R_{eco}$."*

**RC1.5. It would be good to also discuss the results shown by VPRM-SIFg in different season with respect to with previous (conflicting) results in the literature by Gatti et al., (2014), Saleska et al. 2005, Huete et al (2006) (cited in the manuscript) and Brando et al., 2010 (https://www.pnas.org/doi/abs/10.1073/pnas.0908741107).**

We have included additional analysis – namely, we have created a new Figure 10 and compared with the most relevant and recent work by Gatti et al. (2021) (Amazon basin as a whole) and the Cerrado and Caatinga region (also incorporating eddy flux site results from Mendes et al. 2020 and Alves et al. 2021). We have included a summary of these new

results in an additional Section 3.3.3. Comparison with interannual observations. The section and figure are reproduced below. Note that we display the 95% CI in Figure 10b as displaying the larger IQR drowns out the median signal on the plot. Also note that Gatti et al. (2021) report fluxes as Total Flux – Fires ≈ NEE (but not exactly, as the total flux would also include river efflux). However, it's a fair quantity for comparison as NEE dominates that signal.

**3.3.3 Comparison with interannual observations**

*We assessed the performance of VPRM_SIFg and SiB4 from 2010 to 2019 for the Amazon basin (Amazon mask; Fig. 10ab) and separately for the region containing the Cerrado and Caatinga biomes (Cerrado+Caatinga mask; Fig. 10ac) and compared against available observations.*

*For the Amazon mask, the VPRM-SIFg prior tends to estimate interannual net release while the SiB4 model tends to remain closer to neutral (Figure 10b). In addition, the VPRM-SIFg describes greater ecosystem heterogeneity relative to SiB4: the interquartile range (IQR) over the Amazon for the VPRM-SIFg is -0.47 to 0.83 g C m$^{-2}$ d$^{-1}$. In contrast the SiB4 IQR is 0.06 to 0.07 g C m$^{-2}$ d$^{-1}$. Meanwhile, the Gatti et al. (2021) mass balance approach using aircraft vertical profiles tends to estimate net fluxes closer to neutral that generally track SiB4 interannual estimates with a few notable exceptions: in 2016, corresponding to the tail of the severe 2015-2016 El Niño; aircraft profiles suggest a regional net release of 0.1 g C m$^{-2}$ d$^{-1}$ in agreement with VPRM_SIFg, while the following year shows a net regional uptake of -0.2 g C m$^{-2}$ d$^{-1}$. We note that the VPRM_SIFg model agrees with the trajectory of the Gatti et al (2021) post-El Niño fluxes in that there is more net uptake implied between 2016 and 2018. Furthermore, we note that the 2010-2011 El Niño corresponds to a VPRM_SIFg estimate of net release, while Gatti et al. (2021) and SiB4 estimate carbon fluxes that are net neutral to uptake. Given the severity of the associated 2010 drought across the Amazon, particularly as it was only five years after the previous severe drought, it is worth exploring whether the VPRM_SIFg is better able to capture the regional carbon effects and impacts of antecedent environmental stressors.*
*The performance in the Cerrado and Caatinga region suggests that the ecosystem heterogeneity exhibited in the VPRM_SIFg model is realistic. The IQR for the VPRM_SIFg in the Cerrado and Caatinga region captures the site diversity exhibited by the Mendes et al. (2020) northern Caatinga eddy flux site and the Alves et al. (2021) southern Cerrado/converted pasture site. In contrast, the IQR of SiB4 remains closer to neutral. Note that the Gatti et al. (2021) analysis did not include an assessment of the Cerrado and Caatinga regions.*

[Figure]

**Figure 10. Interannual performance of VPRM_SIFg and SiB4 NEE (g C m$^{-2}$ d$^{-1}$) relative to available observations for the decade beginning in 2010.** (a) IGBP land use map overlaid with the Amazon mask, Cerrado + Caatinga mask, and two Cerrado and Caatinga eddy flux sites used for comparison; (b) VPRM_SIFg and SiB4 median annual NEE (95% CI of the median) for the Amazon mask along with estimates from Gatti et al. (2021); (c) VPRM_SIFg and SiB4 median annual NEE (25th, 75th percentiles) for the Cerrado + Caatinga mask along with annual estimates from two eddy flux sites.

We also expanded Sec 3.2.2:

*At ALF the February-March 2016 upwind air masses are potentially significantly influenced by fire activity at all altitudes. In addition, with the upwind air masses bypassing a majority of the Amazon Basin and instead primarily influenced by the Cerrado/Caatinga biome where the models generally agree (Fig. 6d), all prior and posterior flux models perform similarly with a typical model-observation residual of -3 to 3 ppm throughout the vertical column (Fig. 8a–e).*

---

## Author Comment (AC2)

**Response to RC2**

This study focuses on improving the estimation of Amazonian carbon fluxes, particularly the Net Biome Exchange (NBE), which encompasses biogenic and wildfire fluxes. The authors highlight the challenges in quantifying Amazonia's carbon balance due to anthropogenic disturbances and the need for more reliable long-term data. To address this, they utilize solar-induced fluorescence (SIF) data from NASA's OCO-2 satellite and other observations to enhance the Vegetation Photosynthesis and Respiration Model (VPRM). They compare different VPRM versions and the Simple Biosphere 4 (SiB4) model and further optimize these models using OCO-2 $CO_2$ column observations. The study reveals that SIF-based VPRM versions, especially VPRM_SIFg, outperform traditional ones in capturing $CO_2$ fluxes across various timescales and moisture conditions. The researchers underscore the importance of SIF in improving carbon flux estimations and understanding Amazon's response to environmental changes.

We thank Reviewer 2 for the extensive comments, including ways to make both this current work and future follow-on work more robust. Where changes in the paper have been made, comments have been included as "RC2.#".

There are some issues that, if the authors address, can make the study more robust:
**RC2.1   The VPRM model calibration relies on a limited dataset from eight eddy flux sites, potentially hindering the model's ability to represent the diverse and heterogeneous Amazonian ecosystems accurately. (if possible) expanding the calibration dataset to include more sites and diverse vegetation types would improve the model's applicability and robustness.**

We agree but unfortunately, at the time of this work, the eddy flux data set was limited in time and location to the sites as described in the paper. However, this is a topic of ongoing development: in progress VPRM-SIFg development will incorporate newly available data sets representing interior Amazon forest (to supplement K34) and lowland Amazon forest where model-observation mismatch is highest at least as suggested by the March 2016 case study. We now note this in Section 3.2.1 and also in the conclusions:

*Sec 3.2.1: The optimization suggests that all prior models in wet season 2016 tended to underestimate carbon uptake in the western Amazon and particularly in the Amazon lowlands;. The regions of underestimated uptake correspond to the anomalously wet region identified through the Self-calibrated Palmer Drought Severity Index (ScPDSI) by Jimenez-Muñoz et al. (2016), but also in the case of the VPRM to regions under-constrained by eddy flux calibration sites (Fig. 2a; Fig. 6a).*

*Conclusions: Currently, Br-K34 is the only site representing the interior Amazon; additional eddy flux data for calibration and/or validation such as from Amazon Tall Tower Observatory (ATTO) would be beneficial. In addition, eddy flux data from the Amazon lowlands would*

*provide additional constraints to the higher model-observation mismatch observed in that region.*

**RC2.2 The available eddy flux data used for calibration spans a period from 2001 to 2015, which is offset from the study period of 2014-2020. This temporal mismatch might affect the model's accuracy in capturing recent carbon dynamics, as ecosystem states and environmental conditions could have changed over time. Therefore, it becomes essential to incorporate uncertainty in VPRM parameters in inversion runs. These uncertainties can be obtained from the hessian of non-linear optimization procedure.**

Agreed, and we had noted this limitation in the text below. We also fixed the 2014 typo in the paper to 2010 (fluxes are generated from 2010 not 2014)

*Second, while the available eddy flux data can be offset up to thirteen years prior to the study period of 2010–2020, the major drivers of hourly ecosystem flux variations are provided by $T_{air}$, PAR, surface reflectance indices, and SIF. That is, while the static ecosystem parameters of $\lambda$, $PAR_0$, $\alpha$, $\beta$, and $\gamma$ would benefit from tuning to eddy flux data seasonally and/or over the entire study period to reflect concurrent ecosystem states most accurately, the real time variation is dominated by $T_{air}$, PAR, surface reflectance indices, and SIF (Dayalu et al., 2018).*

However, we have also expanded the conclusions accordingly to note the concerns from RC2.2:
*Future work will optimize GPP and $R_{eco}$ separately; in that case, the VPRM can be optimized in parameter space (e.g., Matross et al., 2006) which will also account for the uncertainty associated with using carbon dynamics from 2000-2010 to describe carbon dynamics from 2010-2020.*

**RC2.3 Assessment of transport model errors is required to understand their influence in determining the coherence between observations and fluxes. This can be done through ensemble runs. The authors can also convolve the posterior uncertainty to see the envelope of uncertainty surrounding true observations and convolved observation that is forward operator*posterior fluxes.**

We agree with this and note that a full model optimization (for the entire 2010-2020 time period and for an in-development expansion from 2003-2023) is being conducted. This will include a formal assessment of model transport errors. The scope of the current work was flux model development, with a small performance-testing case study. We had noted as such in Section 2.4:
    *In our application we also neglect transport model errors, which are difficult to quantify but can be further explored in the future using transport ensembles. Errors in WRF-STILT transport are discussed in Rastogi et al., (2021) in terms of calculating $X_{CO2}^{sim}$ for OCO-2 $XCO_2$ retrievals in North America. They found that low partial column bias relative to*

*independent vertical profile CO₂ data show that errors in WRF-STILT transport contribute very minimally to bias in $X_{CO2}^{sim}$.*

**RC2.4   While the study briefly touches upon the impact of fires on carbon fluxes, a more comprehensive evaluation of fire influences, including the effects of fire severity and frequency on ecosystem recovery and carbon balance, would provide a deeper understanding of the Amazon's response to fire disturbances. This can be done by OSSE kind of run to see its impact on the flux.**

Agreed. Once again we note that the scope of this work was to develop a sufficiently representative biogenic flux model that could then be reliable combined with a fire emissions model to complete the picture of impacts on carbon fluxes. The test done here was to find which biogenic flux model provides a realistic representation of the vegetation carbon dynamics (absent confounding influences of fire emissions). In-progress work is developing a CO2 fire emissions inventory from an amazon-specific CO-fire emissions inventory.

**RC2.5   Finally (this is the biggest issue), the study primarily focuses on a wet season case study in 2016 and lacks a long-term validation of the improved VPRM model against independent observations. Conducting a multi-year validation using additional data sources, such as atmospheric CO2 measurements or biomass inventories, would strengthen the confidence in the model's performance and its ability to capture interannual and seasonal carbon trends.**

Agreed, and this was a big gap in the analysis that was also identified by RC1. We have rectified this by including a new section and an additional figure:

We have included additional analysis – namely, we have created a new Figure 10 and compared with the most relevant and recent work by Gatti et al. (2021) (Amazon basin as a whole) and the Cerrado and Caatinga region (also incorporating eddy flux site results from Mendes et al. 2020 and Alves et al. 2021). We have included a summary of these new results in an additional Section 3.3.3. Comparison with interannual observations. The section and figure are reproduced below. Note that we display the 95% CI in Figure 10b as displaying the larger IQR drowns out the median signal on the plot. Also note that Gatti et al. (2021) report fluxes as Total Flux – Fires ≈ NEE (but not exactly, as the total flux would also include river efflux). However, it's a fair quantity for comparison as NEE dominates that signal.

**3.3.3 Comparison with interannual observations**
*We assessed the performance of VPRM_SIFg and SiB4 from 2010 to 2019 for the Amazon basin (Amazon mask; Fig. 10ab) and separately for the region containing the Cerrado and Caatinga biomes (Cerrado+Caatinga mask; Fig. 10ac) and compared against available observations.*

For the Amazon mask, the VPRM-SIFg prior tends to estimate interannual net release while the SiB4 model tends to remain closer to neutral (Figure 10b). In addition, the VPRM-SIFg describes greater ecosystem heterogeneity relative to SiB4: the interquartile range (IQR) over the Amazon for the VPRM-SIFg is -0.47 to 0.83 g C $m^{-2}$ $d^{-1}$. In contrast the SiB4 IQR is 0.06 to 0.07 g C $m^{-2}$ $d^{-1}$. Meanwhile, the Gatti et al. (2021) mass balance approach using aircraft vertical profiles tends to estimate net fluxes closer to neutral that generally track SiB4 interannual estimates with a few notable exceptions: in 2016, corresponding to the tail of the severe 2015-2016 El Niño; aircraft profiles suggest a regional net release of 0.1 g C $m^{-2}$ $d^{-1}$ in agreement with VPRM_SIFg, while the following year shows a net regional uptake of -0.2 g C $m^{-2}$ $d^{-1}$. We note that the VPRM_SIFg model agrees with the trajectory of the Gatti et al (2021) post-El Niño fluxes in that there is more net uptake implied between 2016 and 2018. Furthermore, we note that the 2010-2011 El Niño corresponds to a VPRM_SIFg estimate of net release, while Gatti et al. (2021) and SiB4 estimate carbon fluxes that are net neutral to uptake. Given the severity of the associated 2010 drought across the Amazon, particularly as it was only five years after the previous severe drought, it is worth exploring whether the VPRM_SIFg is better able to capture the regional carbon effects and impacts of antecedent environmental stressors.

The performance in the Cerrado and Caatinga region suggests that the ecosystem heterogeneity exhibited in the VPRM_SIFg model is realistic. The IQR for the VPRM_SIFg in the Cerrado and Caatinga region captures the site diversity exhibited by the Mendes et al. (2020) northern Caatinga eddy flux site and the Alves et al. (2021) southern Cerrado/converted pasture site. In contrast, the IQR of SiB4 remains closer to neutral. Note that the Gatti et al. (2021) analysis did not include an assessment of the Cerrado and Caatinga regions.

[Figure]

**Figure 10. Interannual performance of VPRM_SIFg and SiB4 NEE (g C m⁻² d⁻¹) relative to available observations for the decade beginning in 2010.** (a) IGBP land use map overlaid with the Amazon mask, Cerrado + Caatinga mask, and two Cerrado and Caatinga eddy flux sites used for comparison; (b) VPRM_SIFg and SiB4 median annual NEE (95% CI of the median) for the Amazon mask along with estimates from Gatti et al. (2021); (c) VPRM_SIFg and SiB4 median annual NEE (25th, 75th percentiles) for the Cerrado + Caatinga mask along with annual estimates from two eddy flux sites.

Other comments:

**RC2.6 How are the authors dealing with negative SIF values in their models? If they keep them, GPP will become positive in SIF-based equations. The authors technically are not completely replacing EVI and other scalars by SIF as CSIF is itself derived from MODIS reflectance. It would be good to know how does SIF directly obtained from OCO-2 perform in the VPRM models. Why rely on CSIF when SIF is directly available from OCO-2? Note that OCO-2 SIF also comes with uncertainty, whereas CSIF does not include uncertainty estimates. If there are cloud cover issues, then the distribution of CSIF can be compared with OCO-2. Also, if there are cloud cover issues, then CSIF is mainly influenced by MODIS. I suggest a few things authors can do:**

- **Replace CSIF (derived from OCO-2) with OCO-2 SIF (native OCO-2 SIF)**
- **Compare the ECDF of CSIF with OCO-2. Check if they are similar or not. Use two samples, Anderson-Darling or other statistical tests, to ensure they carry the same information. If they are statistically different, then making any conclusions about SIF improving VPRM estimates would be difficult.**
- **Run this analysis with CSIF, GOSIF, and other SIF products. Please also check the annual variability in CSIF.**

**All this is required to make sure that acceptance of the new model is not an artifact of CSIF.**

OCO SIF was not available prior to 2014, and our study was looking at a decade of biogenic model performance from 2010-2020. For model consistency and to provide the basis for a SIF-based flux model that can be extended to the early 2000s and therefore enable multi-decade carbon trend analyses (this is currently in progress work all the way back to 2003), we wanted a consistent SIF field. The work you mention has already been done in the CSIF paper by Zhang et al. (2018a). Future work will explore a wider range of SIF products, and we have included this statement in the conclusions:
*Future work will continue development of the VPRM_SIFg formulation, including further investigating the model structure as it relates to SIF and PAR as well as exploring the direct use of SIF satellite products rather than derived products such as CSIF.*

**RC2.7 Evaluation of VPRM models against observation is OK, but it depends on uncertainty. Clarify this. Show the posterior flux match of each of them against observations and whether they are within each other uncertainty bounds or they are outside uncertainty bounds, in which case they can be rejected outright.**

Apologies, but we do not understand this comment and were unable to follow-through with a response. To what section are you referring? Are you referring to the optimization? To the aircraft vertical profile calculation? To the eddy flux data?

**RC2.8 The study could benefit from adding a flowchart to provide a clear visual representation of its methodology. This would be particularly helpful in understanding the complex workflow and interconnections between the different components of the study, such as data processing, model calibration, regional inversion, and model evaluation.**

We have created a flowchart and added it to the SI as Figure S4. The flowchart is reproduced below.

[Figure]

[Figure]

**Figure S4. Flowchart of methodology**. (a) Overall methodology; (b) aircraft vertical profile site simulation and comparison.

**RC2.9 Many of the steps the authors took for their assessment need to be formalized to understand better what is being done. For example, "We bootstrap CT2019 background concentrations and vertical profile measurements at each vertical level to dimensions that enable merging with the month of hourly ....." *I was utterly lost here. I do not know what is being done. Line 360 to the end of the methodology section requires a significant rewrite for clarity. Have a clear flowchart + equations + Jupiter notebook. How is this all connected to equation 10. What kind of bootstrap is it?***

We have clarified and re-written Section 2.5, encompassing the area of confusion beginning in Line 360 (reproduced below). Thank you – the section now reads much better. In addition, we have included and equation in the main text for clarity, and also added in a flowchart as a Figure S4 in the supplemental information.

*2.5 Model Aircraft Vertical Profile Simulation*

[revised manuscript text omitted]

**RC2.10 Line 330 "The footprint domain is outlined in Figure Error! Reference source not found.b". Correct this.**
Fixed.

**RC2.11 In Figure 3. VPRM Model-Observation (Night-time) Respiration residuals (I think this is a boxplot). It would also be good to see this as a frequency or histogram plot, as clearly, the bars are not uncertainty estimates. Therefore, we need to know the proportion of residual per/quantile. The authors should explain the relevance of these results in the caption.**

This has been fixed. We have re-displayed the boxplot as violin plots so that both the data distribution and deviation from normal is also apparent (i.e., combining the information in histograms and standard boxplots in one plot). We have also edited the caption and the text to summarize the key take-aways/relevance:

*Sec 3.1:*

*Overall, we find that annually and by season, the SIF-based VPRM formulations—and VPRM-SIFg in particular—have less skewed distributions and lower overall bias than the traditional formulations (Fig. 3, Table S3). On average, the dry season respiration bias is lower across all model versions than in the wet season. In the wet season, while all models tend to underestimate respiration, VPRM-TRA and VPRM-TRG display the greatest bias, with VPRM-TRG displaying the greatest skew. In both seasons, the VPRM-SIFg formulation exhibits the lowest respiration bias with a residual distribution closest to normal. The underestimate in wet season night-time respiration also implies that the general underestimate of peak daytime wet-season drawdown in NEE occurs through underestimating GPP rather than through over-estimating $R_{eco}$. Of the VPRM formulations, the SIF-based formulations have more instances of overestimating respiration, especially in savanna ecosystems (Table S3).*

Associated figure edit:

[Figure]

**Figure 1. Violin plots of VPRM Model-Observation (Night-time) Respiration residuals, annually and by season at eddy flux sites ($\mu$mol $CO_2$ $m^{-2}$ $s^{-1}$).** Nighttime NEE (where GPP = 0) is used to approximate respiration. Lines are 25th, 50th,, and 75th quantiles. All models tend to underestimate wet season respiration and overestimate dry season respiration. Data skew suggests that VPRM_SIFg respiration residuals are the closest to a normal distribution.

**RC2.12 In Figure 4. Panel(a) bars are not uncertainty estimates. They incorrectly imply uncertainty when it is something else. Clarify and, if possible, plot them in a way so that people, by just looking at them, do not think that these are estimates of uncertainty**

Apologies for that; we have clarified that they are the 1-s standard deviation from the NLS fitting of parameters.